# Cutting Processes of Natural Fiber-Reinforced Polymer Composites

**DOI:** 10.3390/polym12061332

**Published:** 2020-06-11

**Authors:** Fathi Masoud, S.M. Sapuan, Mohd Khairol Anuar Mohd Ariffin, Y. Nukman, Emin Bayraktar

**Affiliations:** 1Department of Mechanical and Manufacturing Engineering, University Putra Malaysia, Serdang 43400, Selangor, Malaysia; fathimasoud77@gmail.com (F.M.); khairol@upm.edu.my (M.K.A.M.A.); 2Laboratory of Biocomposite Technology, Institute of Tropical Forestry and Forest Products (INTROP), University Putra Malaysia, 43400 UPM, Serdang 43400, Selangor, Malaysia; 3Department of Engineering Design and Manufacture, University of Malaya, Kuala Lumpur 50603, Malaysia; nukman@um.edu.my; 4School of Mechanical and Manufacturing Engineering, 3, rue Fernand Hainaut, 93400 Saint Ouen, France

**Keywords:** natural fiber, cutting, waterjet, laser beam, roughness, HAZ, kerf

## Abstract

Recently, natural fiber-reinforced polymers (NFRPs) have become important materials in many engineering applications; thus, to employ these materials some final industrial processes are needed, such as cutting, trimming, and drilling. Because of the heterogeneous nature of NFRPs, which differs from homogeneous materials such as metals and polymers, several defects have emerged when processing the NFRPs through traditional cutting methods such as high surface roughness and material damage at cutting zone. In order to overcome these challenges, unconventional cutting methods were considered. Unconventional cutting methods did not take into account the effects of cutting forces, which are the main cause of cutting defects in traditional cutting processes. The most prominent unconventional cutting processes are abrasive waterjet (AWJM) and laser beam (LBM) cutting technologies, which are actually applied for cutting various NFRPs. In this study, previously significant studies on cutting NFRPs by AWJM and LBM are discussed. The surface roughness, kerf taper, and heat-affected zone (HAZ) represent the target output parameters that are influenced and controlled by the input parameters of each process. However, this topic requires further studies on widening the range of material thickness and input parameter values.

## 1. Introduction

For environmental and economic reasons, the production of materials is becoming more sophisticated, more widespread, and more dependent on natural resources [1,2,3,4,5]. Recently, natural fiber-reinforced polymers composites (NFRPs) have attracted many researchers because NFRPs have good properties that can be used in many engineering applications. [2,4,6,7,8]. Although the NFRP composites are manufactured near-net shape, they still need some final cutting processes to reach the shape required such as drilling, cutting, and trimming processes [9,10,11,12,13,14]. Composite materials do not behave like conventional materials under different cutting processes, because of their different properties. Hence, some defects have emerged which require a thorough study of the cutting properties of composite materials to avoid these defects or to reduce their impact [9,15,16,17]. For instance, several defects have arose as a result of the cutting process such as material damage at cutting edges and dimensional instability due to the deformations caused by cutting forces in addition to poor surface roughness [10,18]. In actual fact, the type of cutting process and its variable parameters control the product quality [7]. This has resulted in fundamental questions such as the following: which cutting processes are the most suitable for cutting NFRPs? Which operating parameters are optimal when using a specific cutting process? Cutting processes can be classified into conventional processes that use solid tools such as drilling, sawing and milling, and non-traditional processes which include laser beam machining (LBM), ultrasonic drilling [11], and abrasive water jet machining (AWJM) [19]. The most common methods of cutting NFRPs are covered in this study to answer the questions above on determining which cutting methods are more suitable for cutting NFRP composites and which operating parameters are optimal for each process. It is worth mentioning that the main priority of this study is to cover the cutting processes of NFRPs. Natural composites and other fiber-reinforced polymers FRPs have many similar properties because they are made from the same polymers used in the manufacture of both NFRPs and other FRPs, hence a part of the cutting processes of FRPs is discussed to support this study. FRPs consist of Aramid, glass, and carbon and are used to reinforce polymers.

## 2. NFRP Composites: Definition and Classification

Composite materials consist of two main components, the matrix and reinforcing fibers. In NFRPs composites, the matrix is a type of polymer that binds reinforcing fibers to each other. The matrix is petrochemical based or bio based. NFRPs are classified based on another classification of polymer based into two types, natural fiber-reinforced thermoplastic polymer that can be remolded and recycled, the other type is natural fiber-reinforced thermosetting polymers which cannot be remolded [20]. Reinforcement natural fibers are taken from natural sources, either animal, plant, or mineral [11,21,22]. In accordance to the source, natural and synthetic fibers can be classified as shown in Figure 1.

## 3. Cutting Processes of Natural Fiber-Reinforced Polymer Composites

In this part of the study, the researchers focus on a survey of the most commonly used cutting processes, regardless of the theoretical aspect, which includes many processes that are relatively inefficient to cut NFRP composites such as those that produce low quality or slow production rates. For example, the turning process is not cost-effective because of high tool wear, in addition to poor surface finish [1]. Another example of an inefficient process is ultrasonic machining because it is a very slow process caused by the relatively low material removal rate, and it is relatively expensive and unsuitable for quantitative production [11,12]. Since composite materials are manufactured close to the final shape [9,11], the commonly used cutting processes for making holes, trimming or making profiles are covered in this study. Cutting processes can be classified into two main types: conventional and unconventional cutting processes. Conventional cutting processes include drilling, sawing, and milling [18]. Cutting speed, feed rate, and tool geometry are the controlling parameters in this type of cutting processes whether it is for cutting composite materials or metals [1]. Unconventional methods, on the other hand, include relatively modern processes such as laser beam cutting technology and abrasive water jet cutting process, [11,18] and these methods are the focus of this paper. Both processes have a number of control parameters that are discussed separately in this survey.

### 3.1. Conventional Cutting Processes

Conventional cutting processes of NFRP composites include drilling, sawing, and milling, which are used to produce holes, make profiles and trimming. In conventional cutting processes, the cutting process is accomplished by solid cutting tools, which in turn produces shear forces that separate the material particles from each other. One of the major disadvantages of traditional cutting processes on NFRPs is the production of poor surface quality due to the heterogeneity of the composite material [23]. As a result of shear forces accompanying the cutting process, the phenomenon of fraying damage arises in the case of cutting NFRPs [24]. On the other hand delamination damage occurs in the case of cutting synthetic fiber composites such as carbon and glass fiber composites [18,24,25], while there is no evidence of separation between drilled layers of bio-composites as it occurs in delamination damage [24]. Despite that there are studies that determine that the delamination occurs even in NFRPs cut by conventional cutting process as it is in the studies referred to in the references [11,17,26]. In addition to fraying or delamination deformations, cracking, or smashing is also caused by the cutting force [18,25]. Several studies have been conducted to avoid these defects and efforts have also been made to overcome the effects such as improving the cutting conditions and cutting tools, selecting the appropriate parameters, and controlling the compounds of the material or its manufacturing methods.

A study was conducted by Álvarez et al. [27] on a fully biodegradable composite material which was manufactured from flax fibers and PLA 10361D matrix using the compression molding method. The study focused on changing the drilling point angle of the cutting tool by 6 values, 3 feeds, and 3 cutting speeds with 3 cases of material thickness. From the study it is clear the effect of cutting force that caused different degrees of fraying damage impact. Although noticeable improvements were made in this study, they were still limited because of the huge impact of cutting forces. It is also clear that the cutting tool geometry had a significant effect on the surface quality [28]. H. Rezghi Maleki et al. [29] reported feed rate and drill bit type have significant effect on cutting force with no significant influence of cutting speed was observed on cutting force, and drill bit type has main effect on delamination factor and surface roughness, the study conducted on flax fiber-reinforced epoxy drilled by different type of drill bit with three levels of feed rate and Spindle speed.

In general, the delamination factor and surface damage were significantly affected by the machining parameters [10,27]. For instance, the delamination factor had increased in the presence of both of the cutting parameters (Speed and Feed) as revealed by Venkateshwaran et al. [30]. In contrast, Babu et al. [31] reported that the use of high cutting speed and low feed rate caused minimal delamination, which was agreed by Abilash et al. [26]. Similar to the mechanical properties, the machinability was also affected by the type of fiber [15] and fiber orientation [32].

According to Maleki et al. [17], surface roughness was essentially influenced by the drill bit type, while feed and spindle speed did not show a clear influence on the surface roughness. The material thickness also showed a considerable influence on the process conditions [33].

Natural fiber composites may behave similarly to the synthetic fiber composites under the same cutting conditions. R. Pinzelli [34] reported special tools for cutting composites based on aramid fibers, given the similarity of aramids and ligno-cellulosic fibers are both anisotropic with a tendency to split longitudinally.

Since inconsistent results were found in some of the previous studies that may be due to several reasons such as the variation in fibers, matrixes, manufacturing processes, and the range of input parameters. It is clear that it is difficult to eliminate the defects caused by the cutting forces, hence other unconventional methods must be considered to avoid these defects.

### 3.2. Unconventional Cutting Processes

Unconventional cutting processes are defined as the techniques in which the material is removed by the influence of mechanical, electrical, chemical, or thermal energy or a combination of more than one of these energies, without the use of cutting tools like traditional cutting methods [18,35]. Among the many unconventional cutting methods, only the abrasive water jet cutting process (AWJM) and laser beam cutting technology (LBM) is discussed in this survey.

#### 3.2.1. Abrasive Water Jet Cutting Process (AWJM)

The abrasive water jet mechanism depends on the erosion effect that works on removing the target material using flowing water with high flow rate, speed, and pressure through a narrow nozzle. Hard grains may be added to the flowing water to increase the cutting speed and improve the cutting quality [18,35,36,37,38].

In the abrasive water jet machine, highly pressurized and accelerated water is provided by a lean double acting plunger which is driven by a hydraulic cylinder. The hydraulic system consists of a hydraulic pump which pumps the pressurized oil to drive the plunger. The system is also operated and controlled by a logical system. In the water line, clean and screened water is provided and then it is pressed by the double acting plunger that produces ultra-high pressure and later forced up to 410 MPa through a narrow orifice after being mixed with abrasive grains at the nozzle head. Next, the workpiece is kept at a distance from the nozzle by a specified gap [35,39,40].

There are several important input parameters that determine the quality of AWJM, namely the nozzle or traverse speed, water pressure, abrasive flow rate, type and size of abrasive grains and standoff distance [18,40,41,42]. The output parameters of the AWJM cutting process are surface roughness, material removal rate, and kerf taper angle. These parameters are closely related to the input parameters [18,43,44].

As output parameters, surface roughness, kerf taper angle, and material removal rate will also be the focus of this survey, as well as the phenomenon of delamination, a summary of the relationship between these two and input parameters. It is noteworthy that the AWJM uses erosion to remove the target material, so there is no contact between the cutting tool and the material, and there are no high impact defects of cutting forces related to delamination damage [13,18], but strength of NFRPs machined by AWJM gets affected, so the tensile strength is somewhat lesser than the composites cut by traditional methods [13].

##### Surface Roughness (SR)

One of the recent studies conducted by H.N. Dhakal et al. [19] worked on three types of FRP: flax fiber, carbon fiber, and hybrid flax/carbon reinforced epoxy, with a plate thickness of 2 mm, and the study showed that the surface roughness increased steadily as the traverse speed (*Ts*) increased. Furthermore, the highest value of surface roughness was recorded in flax (natural) fiber-reinforced polymer (FFRP) composite, followed by hybrid flax-carbon fiber-reinforced polymer(C-FFRP), and the lowest surface roughness (*SR*) was carbon fiber-reinforced polymer (CFRP) composites indicated in Figure 2. 

Jagadish et al. [45] also found that *SR* increased with the increase of traverse speed *(Ts*); on the other hand, *SR* decreased with the increase of stand-off-distance (*SoD*) and abrasive grain size (AGS). This study was based on a pineapple leaf filler-reinforced epoxy with 5 mm thickness and the *SR* and water pressure (*Wr*) were found to be not proportional and this is shown in Figure 3. Meanwhile, Jani et al. [16] observed different results of *SR* in the range between 20 to 40 mm/min traverse speed (*Ts)*; for instance, the *SR* reduced with the increase of traverse speed (*Ts*). In this study, Jani et al. [16] also worked on 7 mm thickness hybrid composite that was developed with hemp- aramid (60:40 wt %) reinforced epoxy with and without fillers.

Prabu et al. [46] reported that the *SR* increased with the increase of all of the process parameters: water pressure (*Wp),* traverse Speed *(Ts),* and stand-off-distance *SoD*. The most effective process parameters on the *SR* was the *SoD* with 60.63%, and the study was conducted on a random orientation banana fiber-reinforced unsaturated polyester.

Recently, a similar study was conducted by Kalirasu et al. [47] on two composite glass mat and coconut sheet-reinforced unsaturated polyester with 3 mm thickness. In accordance to the Taguchi’s method analysis, Kalirasu et al. [47] reported that the optimum parameters were: 120 mesh garnet abrasive grain size, 150 MPa water pressure, 3 mm stand-off-distance, and 30 mm/min traverse speed that produced surface roughness and kerf angle.

Most of the previous researchers agreed that there is an direct proportion between the traverse speed and the surface roughness when cutting different types of NFRPs by AWJM, while the results were different in other output parameters. This variation may be due to several factors such as, type of matrix, fibers, method of manufacture, and material thickness.

In this part, some studies related to synthetic fiber-reinforced polymer composites cut by AWJM is discussed in order to achieve a more comprehensive perception of the topic, especially in the light of limited studies that were conducted on cutting NFRPS by AWJM. A recent study was conducted by Kumar et al. [48] who reported that when the water pressure (*Wp*) increased and traverse speed (*Ts*) decreased, surface roughness also decreased and these parameters were most influential on a surface quality of 14 mm thickness plate of aramid epoxy composite cut by AWJM. Meanwhile, Dhanawade et al. [49] who worked on 22 mm thickness carbon epoxy composite workpiece agreed with Kumar et al. [48]. Mullaikodi et al. [50] also agreed with Kumar et al. [48] and Dhanawade et al. [49] but added that the minimum stand-off-distance of about 2 mm to 3 mm must be applied to obtain a minimum surface roughness for cutting 4 mm thickness of hybrid aramid and s-glass fiber-reinforced epoxy by AWJM. From these studies it is clear that surface roughness and traverse speed are directly proportional in case of cutting synthetic fiber-reinforced polymers by AWJM, like the relationship in the case of NFRPs.

A significant variation in the surface geometry and roughness along the cutting depth from the top of the sample to the bottom was seen. The region closest to the top surface was smooth and had less roughness while the surface roughness and waviness increased with the increasing depth toward the bottom of the sample [51,52,53]. Figure 4 illustrates the surface geometry variation along the cutting depth.

##### Kerf Taper Angle (*K*t)

It is one of the most important output parameters of AWJM, which can be defined as a measure of cutting wall deviation compared to the perfect ideal shape [52]. Figure 5 shows a schematic draw of kerf taper generated by AWJM [49]. Similar to the surface roughness, minimizing kerf taper is a major challenge [48,52]. Kerf taper angle can be calculated by Equation (1) [16,42,46,54].
(1)$$Kt=\frac{Wt-Wb}{2t}$$
where *Wt* is the top width, *Wb* is bottom width, and t is the thickness of the sample.

It is hard to form a straight cut because the shape of the jet water after the nozzle may converge or diverge [48]. Equation (2) shows another way to express kerf taper as a proportion of top width of the cut (*Wt*) to the bottom width of the cut (*Wb*) [52].
(2)$$Kt=\frac{Wt}{Wb}\;$$

A recent study was conducted by Müller et al. [55] on a 4.5-mm plate thickness of ground coconut shell fiber-reinforced epoxy composite with and without abrasive. The study evaluated the kerf taper width at the top and bottom of the material plate. The traverse speed (*Ts*) was changed to 50, 100, 250, 500, and 1000 mm/min, and the rest of the process parameters were constant. Figure 6 shows the large influence of abrasive grains on *Wb* especially with 250, 500, 1000 mm/min traverse speed. Since *Wb* did not exceed 1.7 mm in maximum and 1.3 mm in mean, the case used abrasive jet, while the case without abrasive *Wb* reached 0.9 mm in maximum and 0.5 mm in mean value as shown in Figure 7. Figure 6 and Figure 7 also shows that *Ts* had no significant influence at the top cutting zone width (*Wt*), but there is a huge deviation of *Ts* at the bottom of the cutting zone width (*Wb*).

Prabu et al. [46] reported that the reduced water pressure (*Wp*) and the traverse speed (*Ts*) with higher stand-off-distance (*SoD*) resulted in the increase of the kerf angle, and the *SoD* was the most effective parameter followed by the water pressure. It is noticeable that traverse speed did not have a significant effect, contrary to what was stated by Müller et al. [55]. This variation may be due to the difference in matrixes (epoxy versus unsaturated polyester) and the different range of transverse speed. A study on banana fiber-reinforced polyester cut by AWJM was conducted by Kalirasu et al. [54] and the optimum process parameters were: 20 mm/min traverse speed, 2 mm stand-off-distance, and 250 MPa water pressure with 120 mesh garnet grains. Kalirasu et al. [47] also reported that 120 mesh was the optimum size of the abrasive grains but the optimum water pressure (*Wt*) was 150 MPa which was lower in the experiment and the optimum stand-off-distance (S*oD*) was 3 mm, which was the highest value, and this was contrary to all of the above studies. It is clear that there is a variation in input parameters affecting the taper angle, so it is not possible to generalize any study on other NFRP in the event of different fiber, matrix, or the range of operating parameters.

Some of studies are conducted on different synthetic fiber composites. One of these is conducted by Kumar et al. [48] who reported that a minimum kerf angle would be produced with higher *T*s, and this was agreed by El-Hofy et al. [42] who recommended high traverse speed, high water pressure, and small stand-off-distance for a small kerf angle. Dhanawade et al. [49] on the other hand, reported that higher water pressure also produced a smaller kerf taper angle in carbon epoxy composite, but this result can be obtained with high traverse speed and this is shown in Figure 8. It can be concluded that the increase in water pressure and traverse speed produce the lowest taper angle unlike some studies conducted on NFRPs.

##### Material Removal Rate (MRR)

It is a parameter that expresses the amount of material removed by the cutting process relative to the process time, and can be calculated by Equation (3) [45].
(3)$$MRR=\frac{W_{B}-W_{A}}{T}$$

Since *WB* is the processed composite before cutting, *WA* is the processed composite after cutting and T machining time consumption.

Jagadish et al. [45] reported that *MRR* was obviously affected by the process parameters. Their study showed clearly that an increase in stand-off-distance (*SoD)* and traverse speed (*Ts*) led to the increase in MRR. On the other hand, *MRR* decreased with higher abrasive grain size. Also a higher water pressure (*Wt*) resulted in the maximum value of MRR but a minimum value of MRR was obtained with medium water pressure. Figure 9 shows the effect of AWJM process parameters on MRR.

In the case of synthetic fiber composites, Ramesha et al. [41] reported that an increase in stand-off-distance (SoD) water pressure (Wp) led to the increase in MRR, but he had an entirely different view on traverse speed, since there is an inverse relationship between *MRR* and traverse speed (*Ts*). Ramesha et al. [41] added that abrasive concentration as input parameter, could also help to obtain a higher *MRR*, the study conducted on glass fiber-reinforced LY556 epoxy. Meanwhile, Mullaikodi et al. [50] reported that the maximum *MRR* was produced at higher water pressure (*Wp*) and higher abrasive flowrate, but when the traverse speed (*Ts*) was evaluated it was found that the maximum *MRR* was achieved at higher traverse speed (*Ts*) and medium water pressure (*Wp*) for Kevlar-glass-reinforced epoxy.

##### Delamination Studies

In the abrasive water jet cutting technique, the delamination is caused by the effect of the water wedge that made cracks between two adjacent layers of the composite [56]. Most studies conducted on cutting NFRPs with AWJM showed good surface quality without delamination [16,46], unlike the case of cutting synthetic fiber composites by AWJM that exhibited varying degrees of delamination [49,52,56,57]. However, there are limited studies that have shown relatively significant degrees of delamination when cutting NFRPs with AWJM [19,47]. Delamination is similar to the other output parameters and is quite affected by the process input parameters which will be demonstrated in this section of study. H.N. Dhakal et al. [19] worked on tree different types of composites, flax fiber (FFRP), carbon fiber, and hybrid carbon-flax (C-FRP) reinforced epoxy. H.N. Dhakal et al. [19] reported that it was evident that traverse speed (*Ts*) played a significant role in the occurrence of delamination damage, as the delamination clearly increased with the increase of the traverse speed as shown in Figure 10. It is clear that the case of the pure natural fiber composite (FFRP) recorded the lowest value of delamination extent for each traverse speed. In the similar field Kalirasu et al. [47] reported, where the increase of abrasive flow rate had aided noticeably in the occurrence and propagation of delamination, and a significant delamination is seen in the coconut sheath (natural) reinforced unsaturated polyester composite compared to glass mat (synthetic)-reinforced unsaturated polyester composite.

At the end of this section of the abrasive water jet cutting technology, it can be said that there are some of the results from the above studies are incompatible and this is because they were conducted on different thicknesses, materials, and range of parameters.

Despite the importance of material thickness as one of the input parameters which represents the cutting depth of AWJM, it is clear that presently, no studies have covered more than one thickness.

The input parameter selection methodology is unclear in all of the studies covered here, so no clear justifications were given for choosing the specified range of every input control parameter.

However, there is a study conducted by Alberdi et al. [58] on two different types of CFRP with two different thicknesses that focused on the selection of traverse speed by empirical formula for metallic materials. The study aimed to generalize the formula for composites, by finding machinability index for the composites, unfortunately the machinability index of different composite materials was very different, so it was mandatory to define the machinability index separately for every specified material, which did not give this method any important advantage.

Although there are some inconsistencies in results noted in this survey, specifically the results in terms of the effect of the input control parameters on the output parameters, it can be concluded that some parameters have a fixed behavior such as the proportional relation of the traverse speed with the roughness of the surface. Also, the parameters are of varying importance, so it is possible to fix some parameters that do not show a significant impact on the output of the cutting process. Since the AWJM uses water as carrier is there significant uptake of water after machining which may have a significant impact on the results.

One of the most important contributions of AWJM technology is the availability of different drilling methods, including, dynamic piercing, wiggle piercing, stationary piercing, and low-pressure piercing, which showed an important impact on cutting carbon fiber-reinforced epoxy by AWJM [59]. It is important to test this parameter in NFRPs by AWJM. The different piercing methods may explain some of the discrepancies found in the survey.

#### 3.2.2. Laser Beam Cutting Technology

The basic mechanism of the laser beam cutting process can be described as an action of a highly intensive beam of light that is generated by laser with a light wavelength usually in the infrared range. The laser beam is focused on the target material through a lens. Focused laser beam will raise the local temperature of the material to a melting or degradation point, thereby causing local melt or degradation through the depth of the material [60], so thermoplastics are disintegrated by reaching melting point, while thermosetting polymers are degraded directly without melting. The molten or degraded material in the cutting zone is then ejected by a compressed inter gas jet that works with the focused laser beam coaxially, inter compressed gas is also to prevent fire. The cutting zone is later moved across the material surface to generate the cut [61,62,63]. There are many types of laser but to cut the composites, CO2 and neodymium yttrium aluminum garnet (Nd:YAG) are commonly used [11,18,64,65,66].

Laser power, cutting speed, gas pressure, and focal point position are the most important input parameters, in addition to laser wave frequency, type of laser and type of wave (pulsed or continuous wave) [18,61,67,68,69]. These input parameters determine the quality of the cutting process. The main target parameters are: surface roughness, taper angle, heat-affected zone, and kerf width. Hence, the input parameters have to be selected carefully to achieve the optimal values of the target parameters [18,61].

Nugroho et al. [68] who studied how the CO_2_ laser cutting machine had cut the eagle leaf fiber-reinforced unsaturated polyester, reported that gas pressure was the most significant parameter in terms of influencing the kerf taper. Also, the influence of cutting speed, laser power, nozzle distance ranked 2nd, 3rd, and 4th, respectively. 

Eltawahni et al. [70] on the other hand, conducted an investigation study where three thicknesses (4, 6, 9 mm) of MDF boards were cut by the CO_2_ laser. The study focused on the upper and lower kerf width as target parameters, and for a distinctive result, the smallest kerf width was kept precisely at the focal position, so that the smallest kerf width got the lower value when the focal distance was zero, and it gave the optimal surface roughness. In addition, the kerf taper decreased with an increase of the laser power and its effect reduced in thicker materials. The kerf taper angle also increased with the increase of cutting speed up to 3875 mm/min, after that it decreased. Surface roughness meanwhile reduced with higher values of laser power, but its effect also reduced when it was cutting the material. In the study, the surface roughness was clearly becoming poorer with the increase of the cutting speed. During the study, the focal distance was limited between −7 to 0 mm, laser power was between 150 to 600 W, traverse speed was from 2000 to 5000 mm/min and the gas pressure was between 3 to 8 bar.

Therefore, traverse speed, gas pressure, and laser power were the most significant output parameters, meanwhile the other parameters did not show much effect.

It is worth noting that studies on cutting NFRP composites by laser technology are limited and do not provide sufficient reliable data about the input parameters of the process and the extent of the impact of the cutting process on the quality and aesthetics of the edges in terms of the effects of high temperature cutting process by LBM technique, so generalizations cannot be made when cutting other types of NFRP composites because of the limitation of studies conducted on cutting NFRPs using laser cutting technology.

Synthetic fiber-reinforced polymers have had the largest part of studies conducted on cutting composite materials using laser cutting technology. Therefore, to achieve a more comprehensive perception about the topic, some studies conducted on cutting synthetic fiber composites are presented in this survey. One of the recent studies conducted by Raza et al. [71] who focused on the average kerf width and the depth of the heat-affected zone (HAZ) as output parameters in their study and it was conducted on 1.15 mm carbon fiber-reinforced epoxy which was cut by 2 kW Yb-fibre laser (IPG, YLR-2000) machine. Raza et al. [71] reported that both kerf width and heat-affected zone (HAZ) clearly increased with higher values of laser power, and they were inversely proportional to the cutting speed, so high cutting speed gave a good result for kerf width and HAZ, and there were no prominent effects of variation of the gas pressure values. Further, Raza et al. [71] completely agreed with Li et al. [72] about the influence of laser power and cutting speed on kerf width and HAZ, but Li et al. [72] added that the laser wave frequency was an insignificant input parameter. However, the kerf angle had obviously decreased with an increase of frequency in the range of 1000 mm/min of cutting speed and 650 W of laser power, hence the results of kerf angle were quite the opposite at the range of 600 mm/min of cutting speed and 950 W of laser power, but they were less sharp. The study was on 2 mm thickness plate of carbon fiber-reinforced epoxy cut by (IPG YLS-5000) laser machine. Solati et al. [25] found that the cutting speed and laser power had the most effects on HAZ and surface roughness, meanwhile the kerf taper angle was mainly influenced by laser power, for instance higher laser power caused decreasing taper angle. Low cutting speed and high laser power on the other hand, had adverse influence on HAZ and surface roughness. Also, increasing gas pressure contributed to a small increase in HAZ. The study was on 1.2 mm thickness plate of glass fiber-reinforced epoxy cut by CO_2_ laser machine. Next, Herzog et al. [73] conducted a study on carbon fiber-reinforced epoxy that focused on the thickness of material that can be cut by CO_2_ laser machine, and the researchers reported that 6 mm thickness or less could be cut efficiently, and for higher thicknesses, the process lost efficiency but it kept a good kerf width property. According to Li et al. [74] the most effective input parameter on HAZ and surface roughness was fiber orientation, the study was on carbon fiber-reinforced epoxy with 2 mm thickness and three types of fiber orientation: +45°/−45° and 0°/90° respectively, whereas type 3 laminate consisted of four plain woven fiber plies. The results clearly showed that the orientation of the fiber had an important influence on HAZ and surface roughness and +45°/−45° gave the better results than the other ways of orientation. On the other hand the study conducted by Wahab et al. [75] did not show a clear influence of other input parameters such as cutting speed, gas pressure, and laser power, however, the kerf width was influenced negatively with higher values of cutting speed and laser power.

The studies of synthetic fiber-reinforced polymers showed a large variety of results for different materials and cutting conditions, so it is not accurate to rely on these studies for cutting NFRPs by laser cutting technology, especially with the large difference in thermal properties between natural fibers and synthetic fibers.

## 4. Conclusions

The cutting force has a negative impact that is difficult to avoid in traditional cutting, both in delamination and surface roughness. Although several studies have shown that cutting forces decrease as feed rate decreases, but this negatively affects productivity. Delamination is almost non-existent in both unconventional cutting methods, so these methods are the most suitable for cutting composite materials and they have an acceptable range of surface roughness that can be greatly improved if appropriate operating parameters are taken. Another important point that can be gathered from the survey is that there are limited studies conducted on natural fiber-reinforced polymers cut by non-traditional methods especially by laser beam technology. Although the thickness of the material is important as an effective parameter, there have been no studies that cover a satisfactory range of material thicknesses since all of the studies that worked on NFRPs took into account only one thickness. All the researchers also selected a specific range for the values of the input parameters without giving any justifications for these options. A significant discrepancy is observed in the results obtained throughout this survey. For instance, this variation is expected because of the different materials and range of processing parameters used, so it is not accurate to generalize these results to other materials that differ in type, composition, or even manufacturing method; nor can generalization be made for the different ranges of material thickness. Previous studies had shown that in the cutting processes by AWJM and LBM, there were a number of input parameters that had effectively impacted on the properties of the product; while there were other parameters that did not show a significant impact on the output parameters, so these operating parameters had to be chosen and fixed optimally based on those studies. NFRPs are hygroscopic material that may lead to a difference in the output operating parameters because of the difference in the moisture content of the same input parameters, which was not covered by researchers. For all these reasons, more studies have to be conducted in the future for any NFRP composites cut by AWJM or LBM, and these studies should include a satisfactory range of material thicknesses and a suitable range of different input processing parameters to evaluate the desired outputs and determine the most appropriate process to cut the NFRPs.

## Figures and Tables

**Figure 1 polymers-12-01332-f001:**
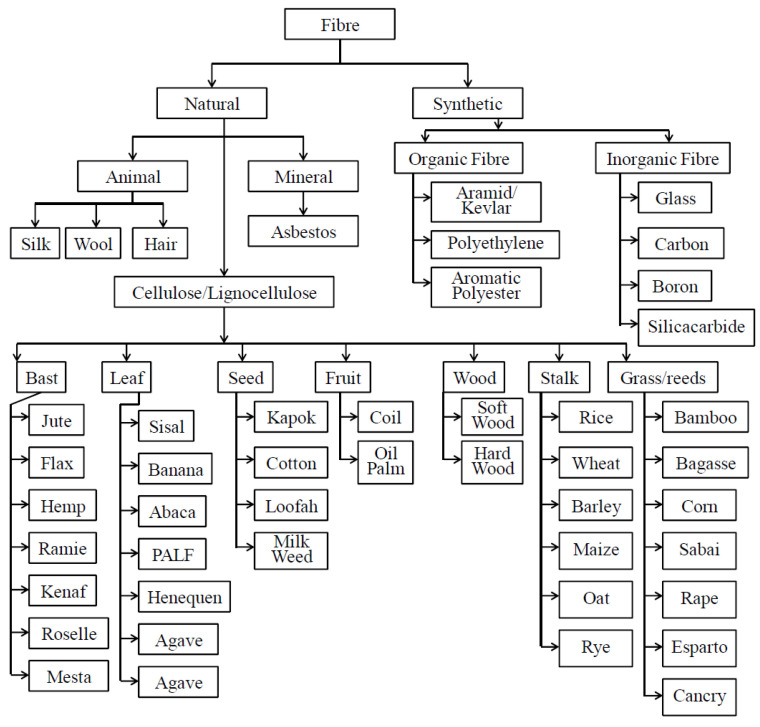
The classification of reinforcement fiber [22].

**Figure 2 polymers-12-01332-f002:**
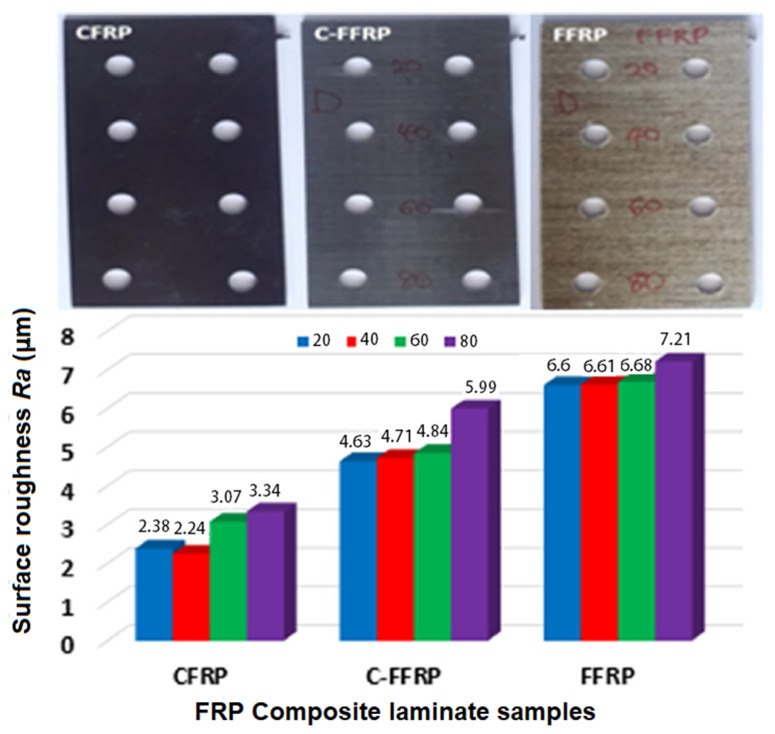
The mean effects of AWJ traverse speed on the surface roughness of sample drilled holes [19]. Image treated to improve quality.

**Figure 3 polymers-12-01332-f003:**
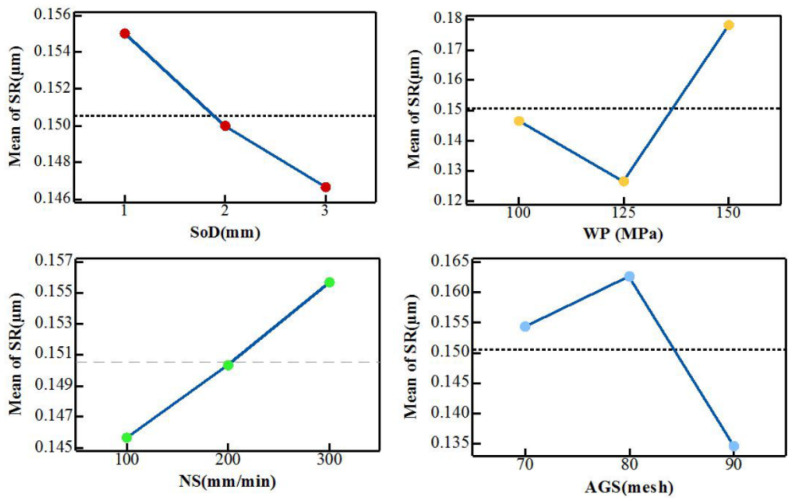
The effects of process parameters on SR [45].

**Figure 4 polymers-12-01332-f004:**
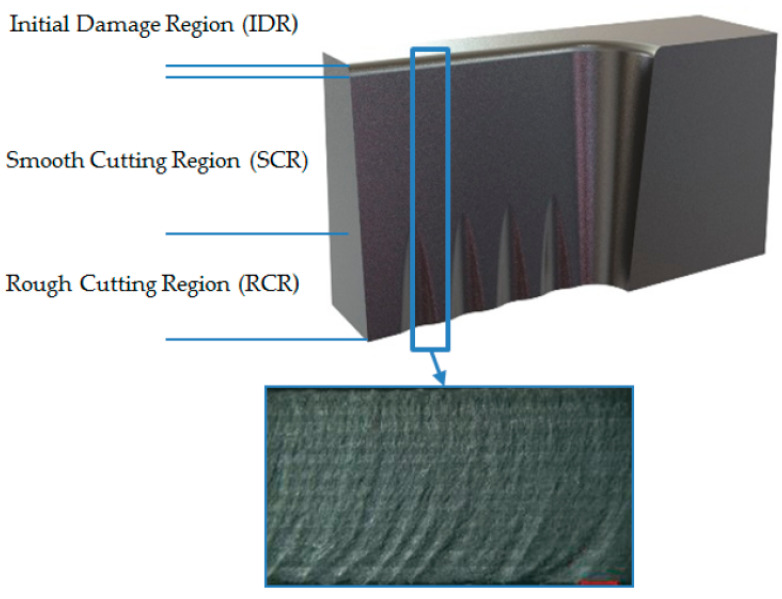
Surface geometry variation along the cutting depth [53].

**Figure 5 polymers-12-01332-f005:**
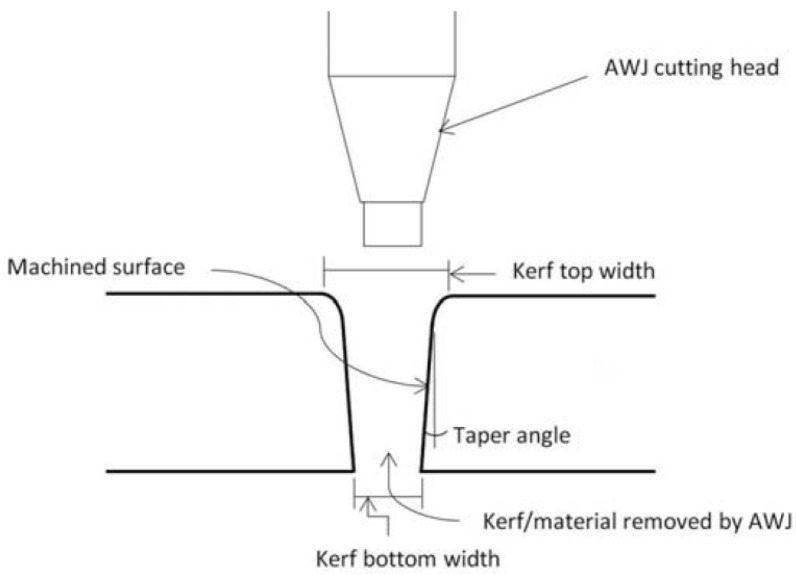
The Kerf Taper produced by AWJM [49].

**Figure 6 polymers-12-01332-f006:**
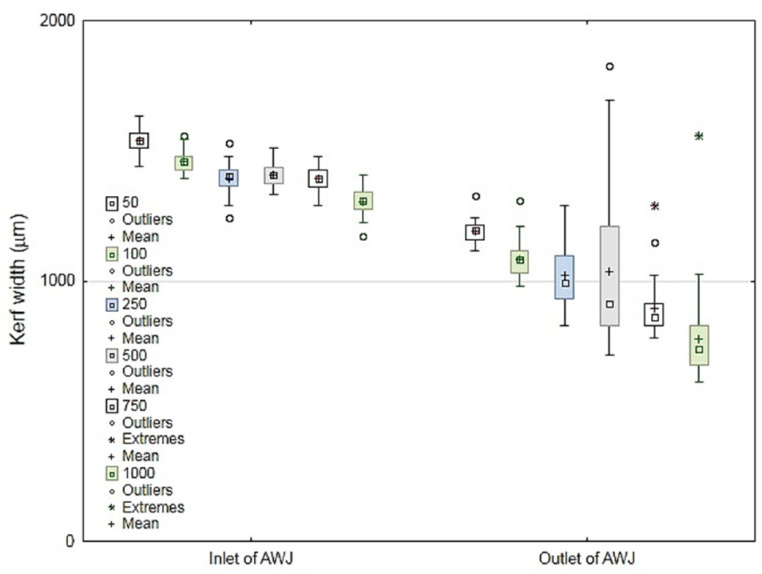
Kerf width of cut with abrasive grains, at the inlet and outlet of cutting zone [55].

**Figure 7 polymers-12-01332-f007:**
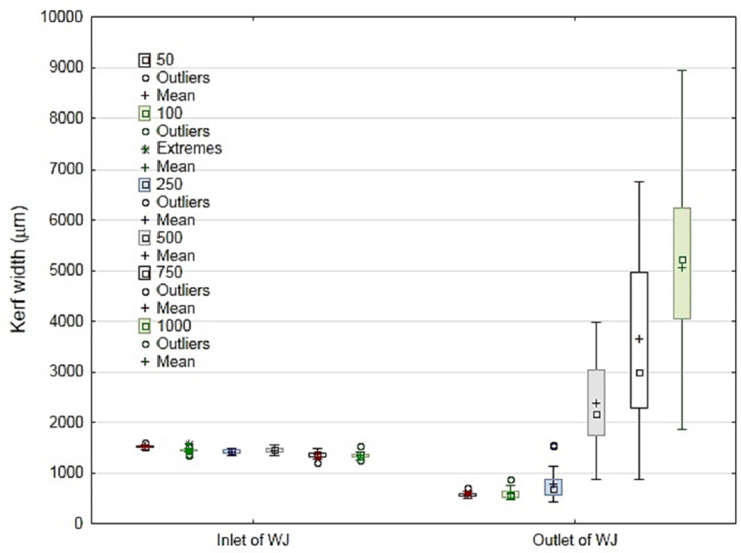
Kerf width of cut without abrasive grains, at the inlet and outlet of cutting zone [55].

**Figure 8 polymers-12-01332-f008:**
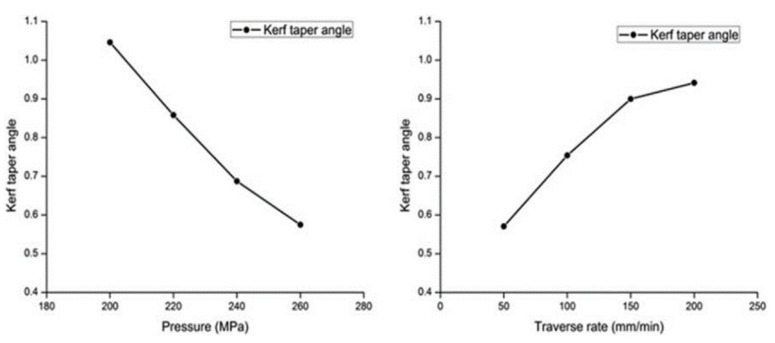
The effects of pressure and traverse rate on Kerf Taper [49].

**Figure 9 polymers-12-01332-f009:**
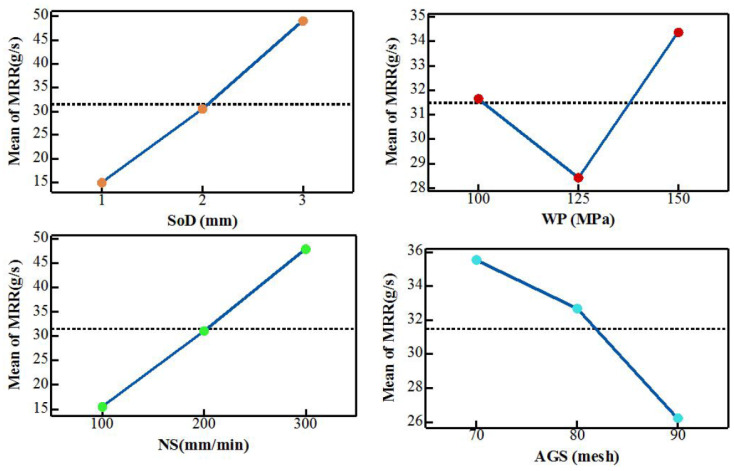
The effects of AWJM process parameters on MRR [45]. In addition, Jagadish et al. [45] and Jani et al. [16] were in full agreement on the influence of water pressure (*Wp*) and traverse speed (*Ts*) which meant that these two were the dominating parameters for *MRR*.

**Figure 10 polymers-12-01332-f010:**
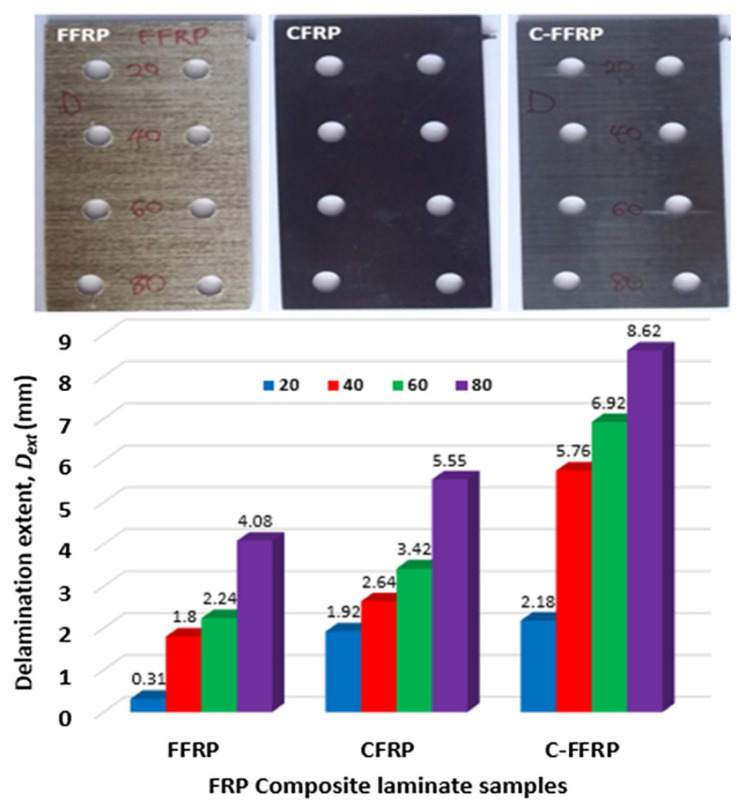
The influence of traverse speed (*Ts*) on delamination damage [19].

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
