# Peer review of "Cutting Processes of Natural Fiber-Reinforced Polymer Composites"

_polymers, 2020, doi:10.3390/polym12061332_

Round 1

Reviewer 1 Report

Comments to your previous version:

The paper is designated “Review” (should perhaps be in the title) with “Natural Fiber” in the title, but significant mention of synthetic fiber in the text!

Natural fibers are hygroscopic, but there is no mention of differences in cutting effectiveness for dry, moist and saturated state NFRP.

Line 58: also mineral natural fibers (asbestos or basalt)?

Line 91: Pinzelli {i and ii} reported special tools for cutting aramids.  Given the similarity of aramids and ligno-cellulosic fibers are both anisotropic with a tendency to split longitudinally, these techniques should be mentioned here? [(i) R Pinzelli, Cutting and machining of composite materials based on aramid fibres, DuPont report H-23157, 1991.  (ii) R Pinzelli, Cutting and machining of composites based on aramid fibres, Composites Plastiques Renforces Fibres de Verre, 1990, 30/4, 17-25. https://www.researchgate.net/publication/284625034_Cutting_and_machining_of_composites_based_on_aramid_fibres].

Lines 131, 176, 236 and 237: capital “P” in “MPa”

Line 132: “gap”, not “gape”

Line 149: why Ra rather than other measurements of roughness?

Line 150 [17]: Dhak[a]l.

Line 153: FFRP is undefined abbreviation!

Line 159: Ts not defined until line 187!

Line 170: Wp not defined until line 186!

Line 194: delete “It is worth mentioning that” as the paragraph would not be present if insignificant!

Line 313: Given the AWJM uses water as carrier is there significant uptake of water after machining?

Line 316: “insensitive”?

Line 318: “melt [or degradation]” for thermosetting resin systems?

Line 319: “inert gas” to prevent fire?

Figure 16 label: “cutting gas”?

Line 340: “eagle fiber” is bird feather?

Line 393: can laser cutting leave a cosmetically nice surface without burn marks?

Reference 31: K.S, no family name?

Reference 32: D.A. no family name?

Reference 48: Initial capital letter of family name for first author?

Reference 49: family name = Mm ?

It may be appropriate to consider the recent review papers on machining of composites before committing this review to publication:

KarataÅŸ and Gökkaya (2018) https://doi.org/10.1016/j.dt.2018.02.001

Ramnath et al (2018) https://doi.org/10.1177/0731684417732840

Vinayagamoorthy (2018) https://doi.org/10.1177/0731684417731530

Vinayagamoorthy and T Rajmohan (2018) https://doi.org/10.1177/0731684418778356

John et al (2019) https://doi.org/10.1177%2F0731684418819822

Kale et al (2020) https://doi.org/10.1016/j.matpr.2020.01.309

Geier et al (2019) https://doi.org/10.1016/j.compositesa.2019.105552

Wan et al (2019) https://doi.org/10.1016/j.compositesa.2018.11.003

Cepero-Mejías et al (2020) https://doi.org/10.1016/j.compstruct.2020.112006

Comments to your new version:

Due to COVID-19 and Working at Home (access to papers in the office forbidden!), I do not have access to the annotated original of my reviewers report.  However, I do recall asking for many of the points below to be addressed!  I compared the original and revised version in Adobe Acrobat Pro DC which suggested there were 634 changes, but that includes over 600 cases of removal of line numbers!  However, none of my proposed enhancements appear to have been addressed!

Two previous reviews (https://doi.org/10.1007/s00170-016-9010-9 and https://doi.org/10.1177/0731684418778356) should, at least, be referenced and perhaps used to inform extension of the paper.

Page 2: “aramid” rather than tradename Kevlar (to include Twaron and other variants!)
Page 2: “NFRP” for consistency, not “NFPC”
Page 4 and page 6: “MPa”, not Mpa, as the unit is named after Pascal!
Page 4: “gap”, not “gape”
Page 4: “Dhakal”, not “Dhakl”
Page 5: define “FFRP” as flax fiber reinforced plastics?
Page 6: Ts introduced before definition some lines later!
Page 12: “insensitive” should be “coherent”
Page 12: “Focused laser beam will raise the local temperature .. to a melting [or degradation] temperature thereby causing local melt [or degradation]”
Page 12: “ejected by a compressed [inert] gas jet”
Page 12 : “eagle” fiber”?
Page 12: “(pulsed or continuous wave)”

I feel certain I flagged the work of Pinzelli at DuPont on new physical-contact cutting tools for aramid fibres (https://www.researchgate.net/publication/284625034_Cutting_and_machining_of_composites_based_on_aramid_fibres) and their structural similarity to natural fibres would suggest those tools would be useful for NFRP?  Further, there is still no discussion of the effect of the water in AWJM on the hydrophilic reinforcement fibres.

New references have been appended to the reference list and high numbers consequently appear out of sequence in the text.

Author Response

Dear Reviewer,

I do not know how can I thank you on behalf of all the co-authors. 

You have helped us too much to improve our paper. 

As you have suggested and critics, we have corrected all of the points. Firstly all of the lines were corrected (58, 91, 131, 132, 149, 150, 153, 159, 170, 176, 194 236 237) and in the same way, we have also corrected references 31, 32, 48, 49.

For the last corrections that you have highlighted were also carried out as you will se in the pages of 2, 4, 5, 6, 12.

As the last version of the manuscript were rearranged
Thank you again for your kind and very useful help.

Yours Sincerely

E.BAYRAKTAR, Full Uni. Professor

Corrsponding author 

Reviewer 2 Report

The paper summarizes the cutting processes used on natural fiber reinforced polymer composites.

Figure 2 shows cutting of carbon fiber composites which is not in line with natural fiber composite, the topic of this manuscript. In addition, fraying instead of delamination is noticed in the graphs. Also Reference 73 is out of order in the text. 

This paper is not suited for publication in Polymers as it is out of the scope of this journal. It is recommended that this paper be submitted to composite-related journals. 

Author Response

Dear Reviewer

Thank you for your timing to review our manuscript. We have revised and corrected the manuscript entirely.

Yours Sincerely 

E. BAYRAKTAR, Full Uni. Professor

Reviewer 3 Report

This work presents a review of the state of art in cutting of natural fibre composites. Authors have shown a deep knowledge of the relevant literature and the main cutting processes are included in this work analysing the most important parameters. However, some issues must be addressed before considering this paper for publication.

  1. In section 2, NFRP are classified in two groups as a function of the matrix: petrochemical or bio based. In this classification authors should mention also the differences between thermoset and thermoplastic because this is relevant for future recycling of the composite.

  1. There a clear unbalance attention to conventional and unconventional cutting processes. For instance, only one figure is included in conventional methods and it is obtained on CFRPs, thus none figures about cutting of NFRP with conventional processes is included.

  1. One of the problems of Figure 2 is that the main failure mechanism is delamination. This is clear in drilling of CFRPs but several authors have reported that delamination was not found in cutting of NFRPs, other failure mechanisms as fraying are more relevant but they are not mentioned in this work.

  1. Authors mention that inconsistent results were found in conventional cutting process. Some of the discrepancies can be attributed to different matrixes and manufacturing processes, these differences lead to different mechanical behaviours, i.e. brittle vs ductile.

  1. Equations are not numbered.

  1. Authors mention different influence of transverse speed in papers [42] and [51]. The reader expect that authors mention possible reasons as the use of different matrixes (polyester versus epoxy) of different range of transverse speed.

  1. In several sections authors compare the results obtained with conventional fibres and natural fibres, but this comparison is not always clear for the reader. For instance, after Figure 10, the results of [37] and [41] are compared but authors do not mention that 41 used natural fibres while [37] use glass fibres.

  1. One of the main contributions of the research on AWJM on conventional composites is the influence of different piercing methods. The different piercing methods can explain some of the discrepancies found in the review of the literature.

  1. In conclusions, authors mention that delamination is almost non-existent. This is true for most of the NFRPs but in the sections of conventional cutting forces delamination is identified as one of the main inducted damages. This is because delamination is the main failure mechanisms in CFRP drilling but fraying is the main failure in NFRP drilling.

  1. In conclusions, authors mention that cutting force is difficult to avoid in traditional cutting. However, several studies have shown that cutting forces increase with feed rate, thus it can be reduced.

Author Response

Dear Reviewer

Thank you so much for your very detailed review of our manuscript. I should explain that I am really very happy for your corrections with very useful and healthy suggestions for our manuscript.

Once again we have used all of the suggestions and critics Under 10 points. And we have revised in view of your critics.

You can find the revised and corrected version of the manuscript here as an attached file

Thank you again

Emin BAYRAKTAR Full Uni. Professor

Round 2

Reviewer 2 Report

The current form of the manuscript is acceptable for me. 

Reviewer 3 Report

Authors have addressed reviewer comments. I recommend this paper for publication in Polymers.